SOFTWARE

# DepoScope: Accurate phage depolymerase annotation and domain delineation using large language models

Robby Concha-Eloko[1], Michiel Stock[2], Bernard De Baets[2], Yves Briers[3], Rafael Sanjuán[1], Pilar Domingo-Calap[1], Dimitri Boeckaerts[2,3]*

1 Institute for Integrative Systems Biology (I²SysBio), Universitat de Valencia-CSIC, Paterna, Spain,
2 KERMIT, Department of Data Analysis and Mathematical Modelling, Ghent University, Ghent, Belgium,
3 Laboratory of Applied Biotechnology, Department of Biotechnology, Ghent University, Ghent, Belgium

* dimi.boeckaerts@gmail.com

**Data Availability Statement:** We provide full access to our code, models and datasets on GitHub (https://github.com/dimiboeckaerts/

## Abstract

Bacteriophages (phages) are viruses that infect bacteria. Many of them produce specific enzymes called depolymerases to break down external polysaccharide structures. Accurate annotation and domain identification of these depolymerases are challenging due to their inherent sequence diversity. Hence, we present DepoScope, a machine learning tool that combines a fine-tuned ESM-2 model with a convolutional neural network to identify depolymerase sequences and their enzymatic domains precisely. To accomplish this, we curated a dataset from the INPHARED phage genome database, created a polysaccharide-degrading domain database, and applied sequential filters to construct a high-quality dataset, which is subsequently used to train DepoScope. Our work is the first approach that combines sequence-level predictions with amino-acid-level predictions for accurate depolymerase detection and functional domain identification. In that way, we believe that DepoScope can greatly enhance our understanding of phage-host interactions at the level of depolymerases.

## Introduction

Bacteriophages (phages in short) are viruses that can infect and kill bacteria. A subset of these phages contains polysaccharide-degrading (PD) enzymes in their virion structure, typically called depolymerases. These play a crucial role in the initial step of the phage replication cycle, *i.e.*, the recognition of specific host surface receptors [1,2]. In nature, bacteria are often found within biofilms. These comprise microbial communities that are enclosed in a matrix of polysaccharides, proteins, nucleic acids and lipids [3]. Phage depolymerases can degrade exopolysaccharides in biofilm matrices, allowing the phage to access secondary receptors [4]. In addition, depolymerases can disrupt the bacterial capsular polysaccharide (CPS), which is an important virulence factor in both Gram-positive and Gram-negative species [5] and can also protect the bacteria from the immune system of its infecting host [6,7]. Despite their importance, functional annotation of phage depolymerases remains challenging. This is specifically

DepoScope) and Zenodo (https://doi.org/10.5281/zenodo.10957073).

**Funding:** MS and BDB received funding from the Flemish Government under the "Onderzoeksprogramma Articiële Intelligentie (AI) Vlaanderen" program (https://www.flandersairesearch.be/en). RS was financially supported by grant GRISOLIAP/2020/158 from the Conselleria d'Innovació, Universitats, Ciència i Societat Digital (Generalitat Valenciana, https://innova.gva.es/va/). PD-C was financially supported by a Ramón y Cajal contract RYC2019-028015-I funded by MCIN/AEI/10.13039/501100011033 (https://www.aei.gob.es), ESF Invest in your future; ESCMID Research Grant 20200063 (https://www.escmid.org/); project PID2020-112835RA-I00 funded by MCIN/AEI /10.13039/501100011033; and project SEJIGENT/2021/014 funded by Conselleria d'Innovació, Universitats, Ciència i Societat Digital (Generalitat Valenciana). DB was financially supported by the Research Foundation – Flanders (FWO, https://www.fwo.be/en/), grant number 1S69520N. The funders had no role in study design, data collection and analysis, decision to publish, or preparation of the manuscript.

**Competing interests:** The authors have declared that no competing interests exist.

due to their high sequence diversity, which is a consequence of the ongoing coevolution between phages and their bacterial hosts [8,9]. A common approach for bacteria to evade infection by phages is to limit access to the receptor by modifying, masking, or even removing the receptor entirely [10,11]. This results in a high diversity of bacterial receptors, which is matched by a diverse set of phage receptor-binding proteins and depolymerases [12]. For this reason, depolymerases are typically difficult to identify using alignment-based methods [13,14]. However, several computational tools and workflows have been developed to facilitate the detection of phage depolymerases in phage genomes. In 2019, Latka *et al.* proposed a manual identification process of phage proteins annotated as tail fiber, tail spike or hypothetical protein using BLASTp, Phyre2, SWISS-MODEL, HMMER and HHPred [15]. The method is based on homology with enzymatic domains and visual recognition of conserved structural features. More recently, two automated, machine-learning-based methods have been proposed. DePP is a Random Forest trained on several experimentally validated depolymerases to discriminate phage depolymerases from other proteins [16]. Similarly, PhageDPO combines a Support Vector Machine and an artificial neural network to detect phage depolymerases [17]. The PhageDPO model was trained on a set of phage depolymerases that were collected based on six depolymerase-associated protein domains. However, a limitation of both tools is the availability of only small or undiversified datasets to train their models (using only a limited number of depolymerase-associated protein domains or only experimentally validated sequences). In addition, beyond predicting a binary outcome for a protein sequence, we aim to precisely identify the functional enzymatic domain in each protein, something none of the current tools can do.

Today, recent advancements in protein language models and protein structure prediction enable us to tackle these outstanding issues in new ways. Protein language models are deep learning-based models that are pre-trained on a large set of protein sequences in a self-supervised way [18]. This means that such models learn to predict which amino acids are present in the context of other amino acids, enabling them to learn general properties and even structural features of protein sequences. Moreover, these pre-trained models can be fine-tuned on a specific prediction task, which implies that what these models learned during pre-training can be transferred and leveraged for other tasks. Typically, fine-tuning only requires relatively few data points and allows making predictions at both the amino acid and the protein levels. Evolutionary Scale Modeling 2 (ESM-2) is a state-of-the-art protein language model developed by Meta AI that is freely available for reuse and for fine-tuning on specific tasks [19,20].

Here, we leverage ESM-2 and the conserved structural features of phage depolymerases to annotate them functionally and characterize their enzymatic domains. We developed a new machine learning tool called *DepoScope* consisting of a fine-tuned ESM-2 model, followed by a set of convolutional and dense neural network layers to both detect depolymerase protein sequences and precisely identify the location of their enzymatic domains. To accomplish this, we collected protein sequences from the INPHARED phage genome database that were processed and filtered both at the sequence and the structure levels. At the sequence level, we screened proteins against a custom database of hidden Markov model (HMM) profiles. These profiles were collected and constructed based on Enzyme Commission (EC) numbers that are associated with a PD activity, which allowed us to capture a great diversity of depolymerases at the sequence level. We screened predicted protein structures against a collection of folds represented in the Carbohydrate-Active Enzymes (CAZy) database with an associated PD activity. A recently developed tool can classify CAZymes broadly but is not specifically tailored towards detecting phage depolymerases [21]. In the CAZy database, enzymes can be categorized into different functional groups spanning glycoside hydrolases, glycosyltransferases, polysaccharide lyases, carbohydrate esterases, auxiliary activities and carbohydrate-binding modules [22]. Of

these categories, glycoside hydrolases and polysaccharide lyases have a carbohydrate-degrading activity. The proteins that passed both screenings were used in a training set for both fine-tuning ESM-2 models to do token classification (*i.e.*, to predict the location of the depolymerase domain) and to construct a convolutional binary classifier on top of the fine-tuned model. This novel approach of collecting data comprehensively and leveraging ESM-2 allows us to identify phage proteins that act as depolymerases and locate the enzymatic domain in these proteins at a high accuracy.

## Design and implementation

**Phage sequence data collection.**   A dataset of phage proteins from the INPHARED phage genome database was collected in January 2023 [23]. This collection comprised 24,289 entries of complete phage genome sequences, of which each protein sequence was pre-annotated via the PHROGS database, a protein resource specifically tailored for phage studies [24]. Protein sequences were then filtered according to their length, any sequence under 200 amino acids in length was discarded from the dataset, which is the minimum length above which we expect to find depolymerase domains [15,25]. A visual overview of the subsequent processing steps is given in Fig 1.

## Generation of a polysaccharide-degrading domains database

To represent the diversity of polysaccharide-degrading enzymes, a comprehensive database of carbohydrate catalytic domains was created. Domain entries from Interpro were fetched in February 2023 [26]. These entries were filtered based on their associated EC numbers. More specifically, those marked with E.C.4.4.2 and E.C.3.2.1, corresponding to carbon-oxygen lyases acting on polysaccharides and glycosidases, respectively, were selected. Further refinement of entries was carried out manually to retain only those for which the association with a carbohydrate-catalyzing activity was based on bibliographic evidence. The final set included 221 Interpro entries (Tab A in S1 Table).

The protein sequences corresponding to each entry were downloaded using an adjusted version of the template script from the Interproscan website. If the catalytic domain was found in an uninterrupted segment of the protein sequence we only extracted that domain. If not, the entire protein sequence was used. Each set of entry-specific sequences was clustered using the MMseqs tool [27], applying parameters "—cov-mode 1—seq-id-mode 1—min-seq-id 0.50 -c 0.8". A multiple sequence alignment (MSA) was generated for each cluster with FAMSA [28], and these alignments were filtered at 95% identity using the HHfilter command from HH-suite3 software [29]. Subsequently, HMM profiles were built based on these filtered MSAs. Lastly, the MSAs and HMM profiles served to construct an HHblits database, completing the formation of the PD domains database.

## Screening the proteins against the HHM profile database and subsequent filtering

MSAs for each sequence were generated from the initial set of proteins in the INPHARED database using MMseqs in conjunction with the UniRef90 database [30]. The clustered sequences were subsequently realigned using Clustal Omega [31], and these final MSAs were screened against the PD domains database using the HHblits command from the HH-suite3 software. Proteins with a bit score higher than 20 with at least 30 aligned positions on their sequence were considered a positive hit. This approach enabled the detection of remote homologies within our MSAs, a valuable asset given the often-high sequence divergence but structural conservation of depolymerase domains.

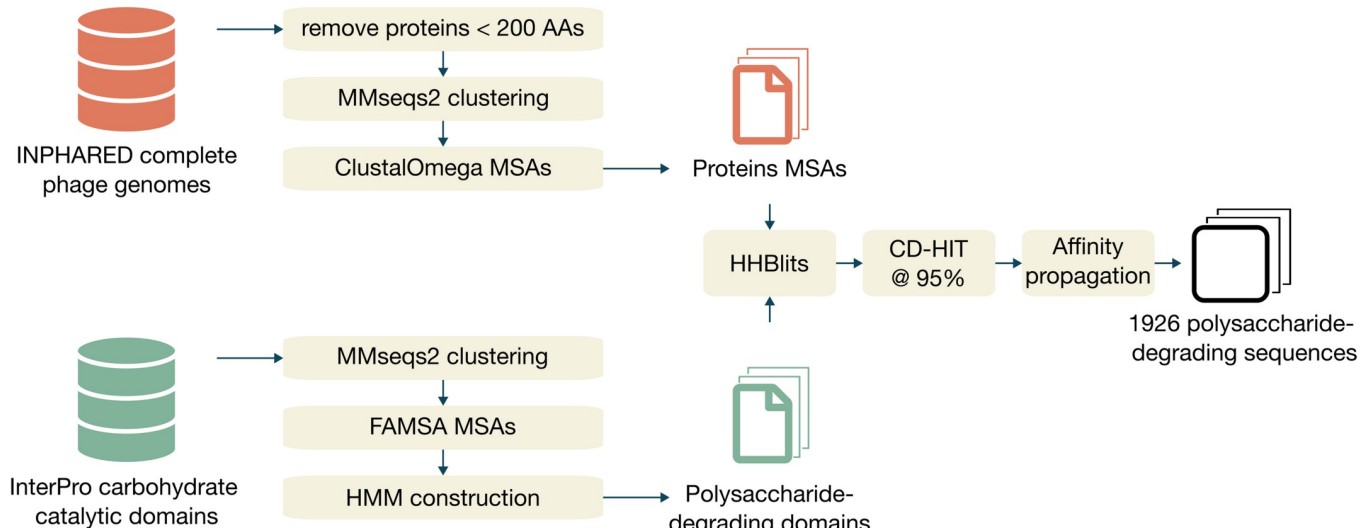

## a. Data collection and sequence-level processing

## b. Structure-level processing

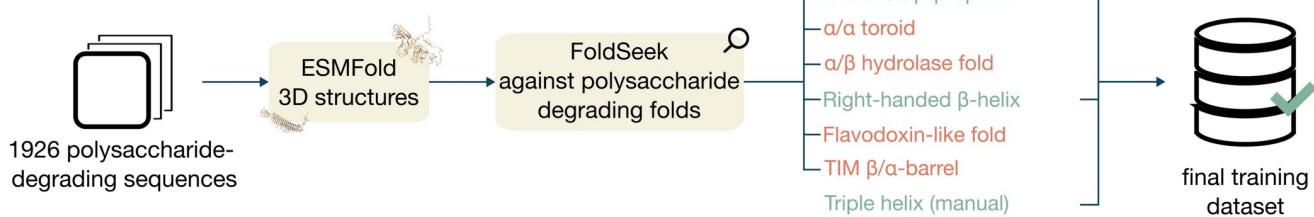

**Fig 1. Overview of the data collection and preprocessing steps at the sequence and the structure level.** (a) Data collection involved collecting raw sequences from INPHARED and InterPro carbohydrate catalytic domains. Protein sequences were processed into multiple sequence alignments (MSAs), while the sequences related to the InterPro domains were used to construct profile Hidden Markov Models (HMMs). Then, HHBlits was used to scan the constructed HMMs against the MSAs, followed by additional clustering and affinity propagation to increase the quality of the collected protein sequence data. (b) The remaining proteins were further filtered at the structure level using ESMFold, FoldSeek and the CAZy database to get to a final training dataset that included sequences with the folds n-bladed β-propeller, right-handed β-helix and triple helix. Icons attributions: Font Awesome Free 5.2.0 by @fontawesome (CC by 4.0); Database by Delapouite (CC by 3.0). No changes were made to the icons.

Sequential processing steps were implemented to reduce the protein count while preserving as much diversity as possible. First, the protein sequences were clustered using CD-HIT [32] at an identity threshold of 95% and an alignment coverage of 80%. The representative of each cluster (*i.e.*, the longest sequence) was kept, narrowing the dataset from 11,513 to 6,816 proteins. Secondly, sequence embeddings were computed with ESM-2 [20] and clustered using affinity propagation [33], employing parameters "damping = 0.90, preference = None", yielding 389 clusters. Finally, we defined a list of annotations consistent with depolymerase activity (Tab B in S1 Table). For each annotation, we included up to five proteins per cluster in the final dataset. This process resulted in a collection of 1,926 proteins.

### Screening the protein 3D structures against the polysaccharide-degrading fold database

The 3D structures of the collection of 1,926 proteins were predicted using ESMFold [20]. A database of domain folds associated with a PD activity (PD fold database) was generated based

on the folds found in the glycoside hydrolase and the polysaccharides lyases families defined by CAZy (Tab C in S1 Table). The folds that were screened against included α/α toroid, right-handed β-helix, TIM β/α-barrel, n-bladed β-propeller (with n > 3), Flavodoxin-like, and the α/β hydrolase folds (Fig 2).

The predicted structures of the 1,926 proteins were screened against this PD fold database using FoldSeek [34]. For each iteration, a hit was considered significant if the associated probability exceeded 0.5. For the right-handed β-helix, hits with a probability that the match is a true positive match greater than 0.2 were considered positive due to the greater divergence of the right-handed β-helix. Finally, a manual screening for proteins with a triple helix domain was done. Triple-helix domains frequently require the assistance of intramolecular chaperones for

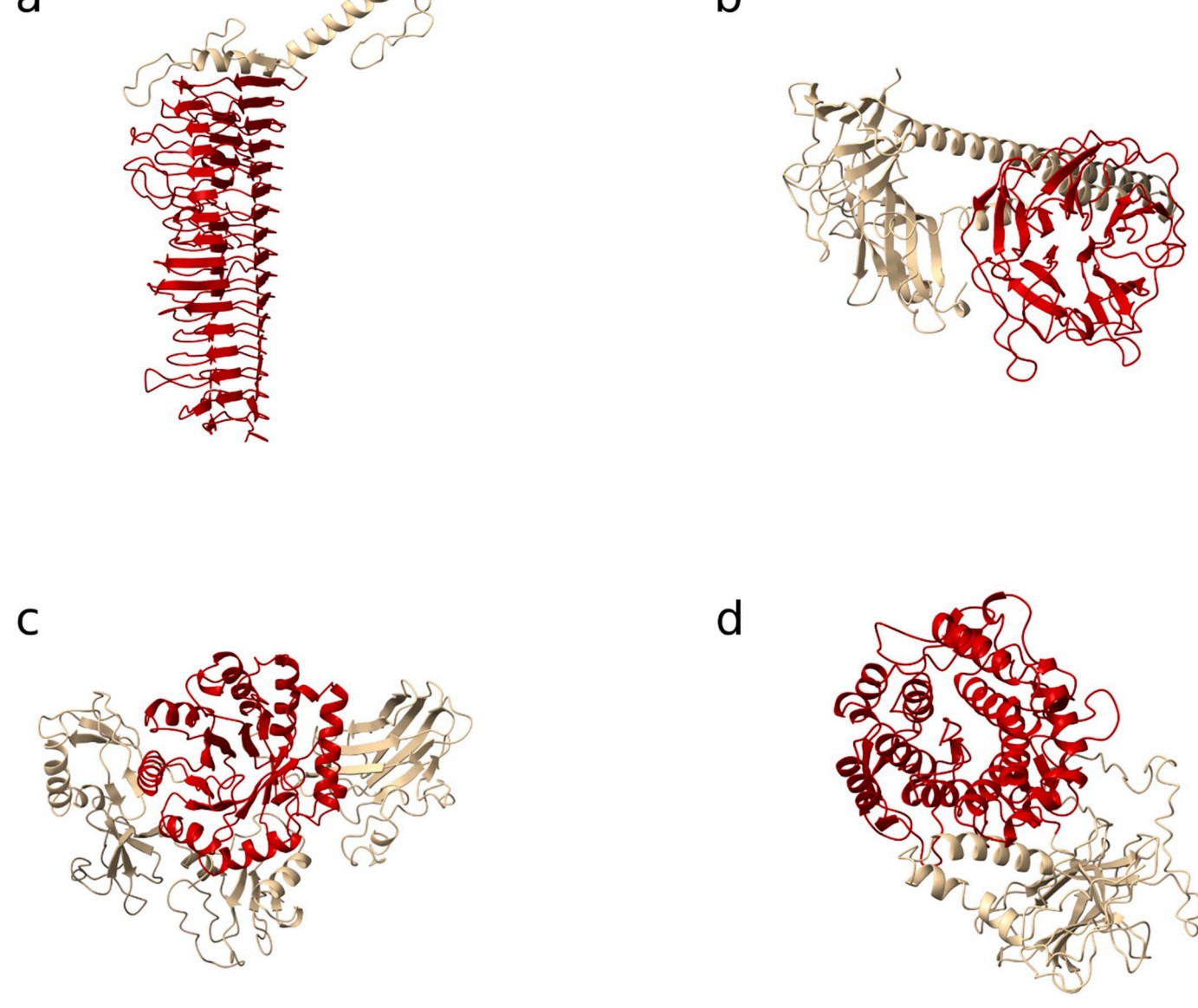

**Fig 2. Identified polysaccharides-degrading folds in the set of proteins from the INPHARED database.** Four different types of PD folds were identified with (a) the right-handed β-helix, (b) the n-bladed β-propeller, (c) TIM β/α-barrel and (d) the α/α toroid. The PD fold is colored in red, while the remaining part of the protein was left in gold.

their accurate and stable folding [35,36]. This reliance highlights the intrinsic instability characteristic of these domains, often manifesting as structural disorder [37,38]. This instability engages in a multifaceted interplay with additional variables, such as the pre-chaperone activity disorder in the 3D conformation and the predictive capabilities of sequence-based models to accurately determine the protein's 3D structure. These complexities introduce challenges to searching for reliable structural similarities between those folds.

## Delineation of depolymerase domains

The 3D structure predictions of the proteins that demonstrated a significant hit with the PD fold database were used to identify corresponding (so-called) protein units: stable segments within a protein that fall between secondary structures and domains in terms of their complexity. These were determined with SWORD2, which analyzes the spatial relationships between α-carbon atoms and optimizes for structural autonomy within these segments [39]. Every protein unit identified was subsequently screened against the PD fold database to locate the best match. The delineation of the PD folds was inferred from the best match. These processes resulted in a final dataset of 602 right-handed β-helix, 96 n-bladed β-propeller and 146 triple-helix depolymerase proteins for which the catalytic domain was precisely determined. As an exploratory step, feature representations of the delineated PD folds were computed with ESM-2 and plotted using t-SNE (S1 Fig).

## Model construction and evaluation

DepoScope is built as a stacked model architecture with two parts (Fig 3). The first part is a fine-tuned ESM-2 model for a token classification and the second part is a pair of two convolutional layers and two forward neural layers for a binary classification task. Both modules have been trained sequentially, first considering the token classification task followed by the binary classification.

For training and evaluation, a set of protein sequences that do not match any of the described PD domains above were added to the dataset as 'negative' sequence instances for the model. We specifically chose to collect other proteins that are involved in catalytic activity and structural proteins without catalytic activity, as those proteins would be potentially hard for the model to correctly classify as negatives (Tab D in S1 Table). The full training dataset was split into three parts: 70% for the token classification task, 20% for the binary classification task and 10% for the evaluation of both tasks.

In this study, the pretrained ESM-2 models (esm2_t6_8M_UR50D, esm2_t12_35-M_UR50D, esm2_t30_150M_UR50D configurations) were used to construct fine-tuned models for a token classification task. Each amino acid of a given sequence was classified into a class corresponding to the different PD domain folds that the model is being trained on. The model was fine-tuned using the Python libraries Transformers and Pytorch and optimized the model hyperparameters via Bayesian search with the Optuna package [40].

In addition, a second deep learning model on top of the ESM-2 fine-tuned token classification model was trained for a binary classification task (prediction score threshold of 0.5). Two convolutional layers identify relevant patterns in the list of labels assigned to each amino acid. These patterns are then processed by two forward neural network layers that output a prediction score reflecting how likely the protein sequence is a depolymerase. The final outputs of DepoScope are the list of labels for each amino acid and the probability associated with the input sequence being a depolymerase. In order to validate the ability of the model to generalize, the Group Shuffle Split method was used to split the data. Specifically, the sequences were clustered with CD-HIT before training using a range of clustering values (0.25, 0.30, 0.35, 0.40,

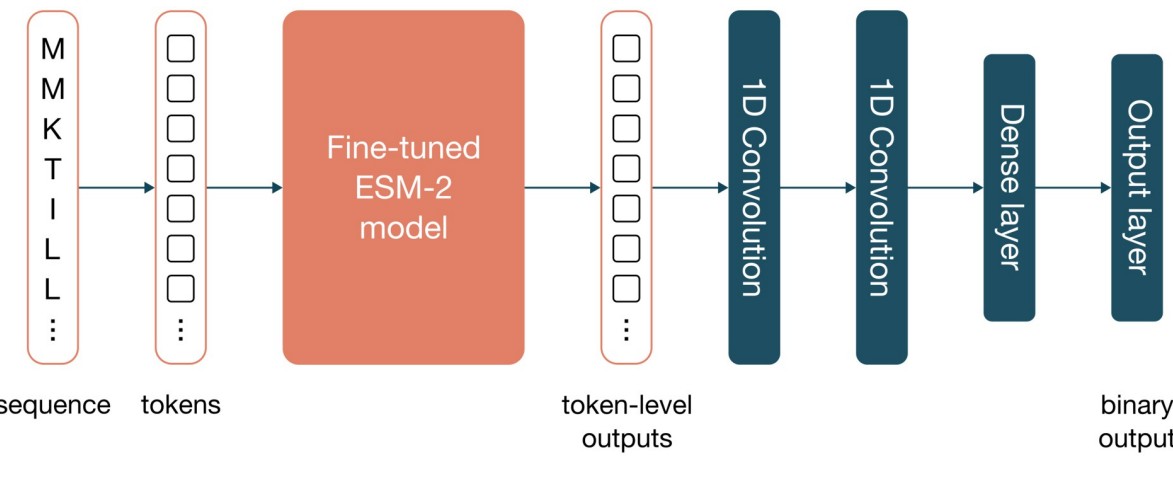

**Fig 3. DepoScope model architecture.** DepoScope consists of a combination of two deep learning models that perform token classification and binary classification, respectively. The token classification model is a fine-tuned ESM-2 model that receives protein sequences as input, which are transformed into tokens (one for each amino acid). For each token, the model learns to classify it as being part of a PD domain or not in the finetuning process, with four distinct labels: "none", "right-handed β-helix", "n-bladed β-propeller" and "triple helix". The outputs of this first model additionally serve as inputs to the second model, which is a combination of two convolutional layers and a dense layer that produce a binary output, which reflects the prediction for the entire sequence of whether or not the protein is a depolymerase.

0.45, 0.65, 0.70, 0.75, 0.80, 0.85). No significant impact on the performances was observed, indicating an ability to generalize on unseen data (Tab E in S1 Table).

## Benchmarking against available tools

The DepoScope machine learning model was benchmarked for binary classification against two available tools for depolymerase detection: DePP and PhageDPO [16,17]. The source code of both tools was downloaded from their respective repositories on GitHub or Galaxy, and we ran both tools locally in an iPython notebook. As an evaluation dataset, the depolymerase dataset included in Pires *et al.* (2016) was chosen, which was also used by the authors of DePP for benchmarking [25]. More specifically, the protein sequences (> 200 amino acids) of all 142 phage genomes in the Pires dataset were collected. To mitigate biases, sequences present in the training dataset of either of the tools were not considered. We asked each tool (including DepoScope) to make predictions for each of the protein sequences across all phage genomes. Based on these predictions, together with the true labels of each sequence (as given by Pires *et al.* [25]), the recall (TP / [TP + FN]), precision (TP / [TP + FP]), F1 score ([precision + recall] / [2 * precision * recall]), specificity (TN / [TN + FP]) and Matthews Correlation Coefficient (MCC; [TN*TP-FN*FP] / sqrt[(TP+FP)*(TP+FN)*(TN+FP)*(TN+FN)]) were computed to compare the three tools across a range of performance metrics.

## Results

### Dataset construction and processing

We initially collected 554,981 proteins from the INPHARED database (consisting of complete phage genomes) that were longer than 200 amino acids. These proteins were further processed

in multiple steps. A total of 11,513 proteins presented a significant hit against a custom PD domain database (comprising sequences across 221 InterPro entries). Their 3D structure predictions were scanned against the PD fold database. A set of 984 proteins presented a significant hit with a fold from the PD fold database (Fig 1a). Most of the detected folds were right-handed β-helices (844 times, 85.8%) and n-bladed β-propellers (119 times, 12.1%). Additionally, 11 hits (1.1%) and 10 hits (1.0%) were identified as α/α toroid and TIM β/α-barrel folds, respectively (Fig 2). We were then able to pinpoint the exact location of the PD domain in 602 of the proteins harboring a right-handed β-helix and 96 out of those carrying an n-bladed β-propeller.

We annotated 146 triple helix domains manually and added these to the training set (Tab F in S1 Table). In summary, the training of our models included the right-handed β-helix, the n-bladed β-propeller and the triple helix folds. The TIM β/α-barrel and α/α toroid folds were eventually not included in the training process of the model due to the low number of identified instances. For every filtered protein sequence, each amino acid was labeled according to whether it was part of a depolymerase domain and with the type of fold, resulting in four labels: "none", "right-handed β-helix", "n-bladed β-propeller" and "triple helix".

Finally, negative sequences were added to the training dataset. In total, 1,409 proteins were collected that matched the annotations of interest and these were added to the training dataset as negative instances. As the number of sequences in the positive and negative classes do not differ substantially, we have not considered artificially balancing the training dataset.

## Training the model

The three different ESM-2 model configurations for finetuning were compared based on their performances on the evaluation dataset and their computational requirements. As expected, the ESM-2 30L had the highest MCC score of 0.903, followed by ESM-2 12L and ESM-2 6L with MCC scores of 0.898 and 0.884, respectively (Table 1). The fine-tuned ESM-2 models most often confused one of the PD domain tokens with the negative label token (not with other PD domain tokens) while maintaining the level of confusion relatively low (Fig 4). For the binary classification task, DepoScope reaches an MCC score of 0.987, 0.975 and 0.958 for the ESM-2 30L, ESM-2 12L and ESM-2 6L configurations, respectively (Table 1). The ESM-2 12L mode configuration was kept as the final model because of the balance between the memory requirement and the model efficiency.

## Benchmark

To further evaluate DepoScope, we benchmarked it against two recently developed depolymerase detection tools, DePP and PhageDPO, on the phage depolymerase dataset published by Pires *et al.* (2016, Tab G in S1 Table) [25]. We computed six performance metrics (at a

**Table 1. Performances of the fine-tuned models ESM-2 6L, ESM-2 12L and ESM-2 30L for the token classification task on the evaluation dataset (best results annotated in bold).**

| Task | Model size | precision | recall | accuracy | f1 | MCC | Running time (sec / 100 k amino acids scanned) | Memory consumption (KB) |
|------|-----------|-----------|--------|----------|-----|-----|-----------------------------------------------|-------------------------|
| Token classification | ESM-2 6L | 95.4% | 95.4% | 95.4% | 95.4% | 0.884 | - | - |
| | ESM-2 12L | 96.0% | 96.0% | 96.0% | 96.0% | 0.898 | - | - |
| | ESM-2 30L | **96.1%** | **96.1%** | **96.1%** | **96.1%** | **0.903** | - | - |
| Binary classification | ESM-2 6L | 99.5% | 95.6% | 98.0% | 97.5% | 0.958 | **32.4** | **331.34** |
| | ESM-2 12L | **100%** | 97.1% | 98.8% | 98.5% | 0.975 | 89.7 | 600.11 |
| | ESM-2 30L | **100%** | **98.5%** | **99.4%** | **99.3%** | **0.987** | 229.5 | 1769.26 |

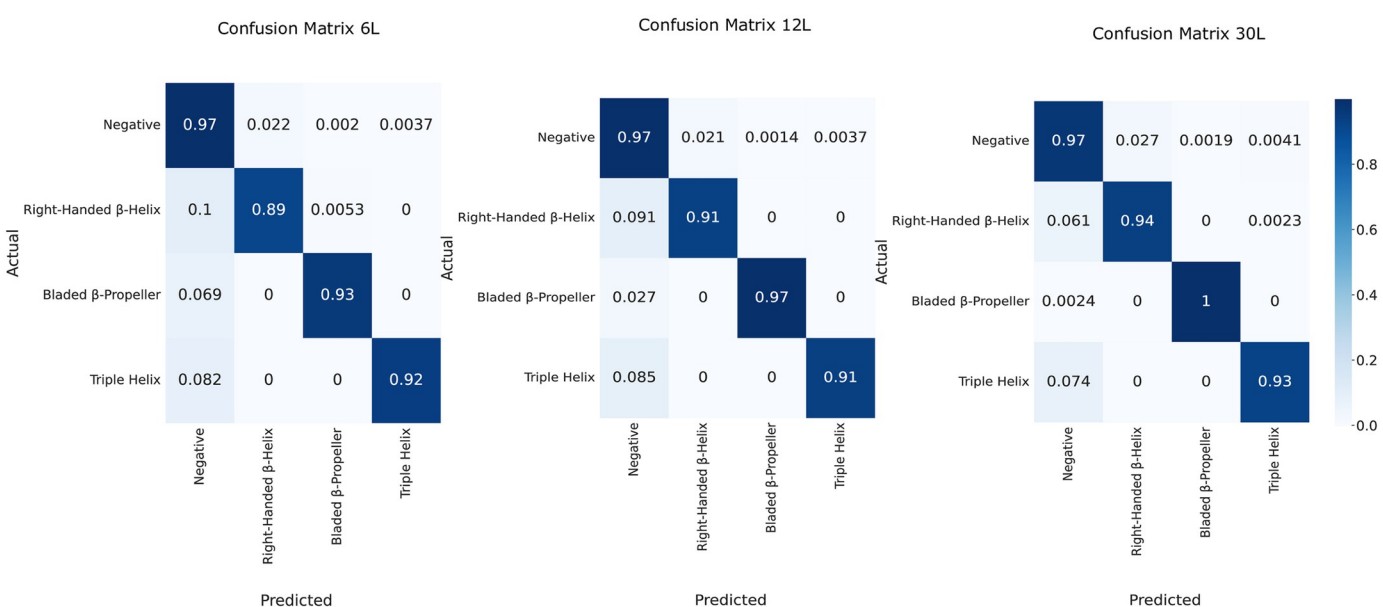

**Fig 4. Confusion matrices of the token classification task for the T6, T12 and T30 configurations of the fine-tuned ESM-2 model.** Four labels were predicted by the models with "none", "right-handed β-helix", "bladed β-propeller" and "triple helix".

prediction threshold of 0.5 when relevant): precision (P), recall (R), accuracy, F1 score, MCC and the PR area-under-the-curve (Table 2). DepoScope outperforms both other tools on all metrics except for recall. Moreover, reaching an MCC of 0.455, DepoScope was more than twice as performant compared to the other tools, reaching MCC scores of 0.131 and 0.178 for DePP and PhageDPO, respectively.

This difference is largely explained by DepoScope's ability to minimize false positive (FP) predictions, having a total of 123 FPs versus 2103 and 966 for DePP and PhageDPO, respectively. The recall was highest for DePP with 91.6%, while DepoScope and PhageDPO reached a recall of 69% and 74.7%, respectively. Conversely, DePP and PhageDPO achieve precisions of 3.5% and 6%, respectively, while DepoScope maintains a precision of 32%. Overall, DepoScope achieves a PR AUC of 42.3%, compared to 10.2% and 27.8% for DePP and PhageDPO, respectively. Most of the false negative predictions for DepoScope corresponded to the triple helix fold (18 out of 26), followed by α/β hydrolase (4 out of 26), the TIM β/α-barrel (2 out of 26) and the n-bladed β-propeller (2 out of 26). The performance of DepoScope was further assessed by analyzing the FP predictions. For each of the 123 FPs, the 3D structure was predicted using ESMFold and then scanned against the PD fold database. Out of the 123 FPs, 78 had a significant hit against the PD fold database, with 41 right-handed β-helix and 37 n-bladed β-propeller. Manual investigations of the 45 remaining FPs revealed 32 proteins harboring a PD fold. These findings suggest that Pires *et al.* [25] have missed some depolymerases

**Table 2. Benchmark results for the three tested models.**

| Models | precision | recall | specificity | accuracy | F1 | MCC | PR AUC |
|---|---|---|---|---|---|---|---|
| DePP | 3.5% | **91.6%** | 60.9% | 61.4% | 6.7% | 0.131 | 10.2% |
| PhageDPO | 6.0% | 74.7% | 82.0% | 81.9% | 11.2% | 0.178 | 27.8% |
| DepoScope | **32.0%** | 69.0% | **98.0%** | **97.2%** | **43.4%** | **0.455** | **42.3%** |
| DepoScope FP adjusted | 81.4% | 69.0% | 99.8% | 99.3% | 74.5% | 0.744 | 51.6% |

in the phage genomes (Tab H in S1 Table and S1 Data). For the remaining 13 FPs, no indications were found that point to depolymerase activity. Those 13 FPs were annotated as hypothetical or structural proteins such as tail protein, tail fiber protein, tail sheath and baseplate proteins. When only considering the remaining 13 proteins as FPs, the MCC of DepoScope increases to 0.744 (referred to as Deposcope FP adjusted).

## Availability and future directions

We developed DepoScope, a language-model-based tool that can both detect depolymerase protein sequences and accurately identify the amino acids involved in their enzymatic domains. We firmly believe that open research is valuable for the field to keep moving forward swiftly. For this reason, we provide full access to our code, models and datasets on GitHub (https://github.com/dimiboeckaerts/DepoScope) and Zenodo (https://doi.org/10.5281/zenodo.10957073).

Our work goes beyond the current state-of-the-art in depolymerase annotation by approaching data collection and processing in a novel and comprehensive way, followed by leveraging the protein language model ESM-2 [20]. Today, various protein language models (including ESM-2) are open-source and can be easily fine-tuned and adapted for downstream predictive tasks. Moreover, protein language models are an area of active development, which can result in future improvements of DepoScope as well. Specifically, our tool can be refined at three different stages: the pretraining data collection, the pretraining model architecture and the fine-tuning data. This is different from classical machine learning models, which do not leverage a pretrained model and can only be improved in terms of feature engineering (equivalent to improving the model architecture of deep learning models) or training data. In addition, the transformer-based architecture of such models naturally allows for predictions at the level of individual tokens (here, amino acids). Such an approach has been used previously for predicting protein secondary structure [19,41], and here we use it for amino acid level annotation. We believe that predictions at this level can give a refined look at what exactly is being predicted by the model, especially when predictions are further investigated using 3D structure predictions. Consequently, such detailed predictions can improve and simplify depolymerase engineering workflows, which typically require multiple manual analysis steps [2]. On the contrary, making amino acid level predictions brings forth the additional challenge of establishing reliable labels at the level of individual amino acids, which is not straightforward given the limited number of experimentally verified depolymerase protein structures. Here, we overcame this challenge by implementing an elaborate data processing pipeline that included (1) the SWORD2 algorithm for delineating protein units, (2) a set of specific PD folds from the CAZy database and (3) the FoldSeek tool to screen matching protein domains of interest. However, we believe that this processing pipeline can be further improved (e.g., by making more explicit use of experimentally determined structures and manually delineating their domains).

From a broader perspective, it is apparent that the typical way in which researchers functionally annotate proteins is changing nowadays. Traditional methods, such as using alignment to detect homology or using profile HMMs as quantitative representations of functional domains, are being complemented or replaced entirely with deep-learning-based models for annotation. A notable example is the work by Bileshi *et al.* [42], in which they use an ensemble of convolutional neural networks to annotate protein domains, which, combined with existing methods, led to a significant expansion of the Pfam database. We believe that sophisticated deep learning methods are indeed proving to be excellent complementary tools for protein functional annotation and will continue to be in the future. However, this does not diminish

the importance of experimental verification of protein function and manual curation of protein sequence data and annotations.

Compared to other tools for depolymerase annotation, DepoScope is more performant across every computed metric except for recall. The reasons for this result are the limited number of data points (DePP) or the limited diversity in the underlying folds to collect data with (PhageDPO). Both DePP and PhageDPO result in many false positive predictions on the benchmark dataset by Pires et al. [25]. The authors of DePP mitigate this by using a ranking approach. We argue that this is not a particularly useful output format because it prevents making a single annotation prediction for each protein separately and hence necessitates comparing prediction scores across multiple proteins. In addition, our method also results in some false positive predictions, according to Pires. However, looking at the 3D structures of these proteins revealed that many of them match against PD domain folds in our training set, which could indicate that some depolymerases were missed by Pires et al. in analyzing their phage genomes [25].

DepoScope has been trained to predict the presence of a selection of folds, including the β-helix and the n-bladed β-propeller. Those conformations were the most prevalent in the folds detected by our method, with the β-helix being by far the most abundant one. The potential biochemical advantages of the β-helix could explain this unequal distribution. Firstly, the stability of this conformation has been widely described [43]. This could permit the phage to actively infect hosts in different ecosystems. Secondly, the nature of the β-helix allows for a vast diversity in its length and sequence [44]. These features could benefit the phage in the face of the evolutionary arms race with the bacteria. The capsule is subject to rapid changes or swaps that put evolutionary pressures on the phage to adapt [8].

Several uncommon folds have been observed in the collected phage proteins with a PD fold. To our knowledge, some of these folds have never been studied in the realm of viruses, specifically in phages. Further investigation on the depolymerase activity in phages carrying those folds could provide valuable insights into the activity and ecology of phages. Some of those folds might hold interesting properties. Previous studies reported a substrate specificity depending on the pH, which implies important ecological and technological implications [45].

Our tool succeeded at being highly performant in predicting the β-helix and the n-bladed β-propeller in the phage genomes presented by Pires et al. [25], but not as much in predicting the triple helix, which consisted most of the false negatives. We see two reasons that could explain these errors. Firstly, the level of disorder in the triple helix domain (before trimerization), which only decreases after the activity of the chaperone [35,36]. This high level of disorder can potentially make the pattern more complex and hampers proper prediction based only on the sequence. Theoretically, this could be bypassed by adding more divergent sequences in the training data and/or using a bigger pre-trained ESM-2 model, which can understand more complex relationships between amino acids, resulting in a higher sensitivity for more complex patterns. Secondly, the depolymerase activity in those proteins can be harder to assess. Indeed, in most cases, the depolymerase activity has been deduced from the presence of the chaperone domain, which could potentially lead to false positives.

In conclusion, our study fundamentally rethinks the traditional understanding of depolymerases by offering a more accurate identification, including delineating the enzymatic domain. We suggest that depolymerases should be defined as phage proteins with an enzymatic PD domain with a specific fold known for their ability to degrade polysaccharides. These insights have broader implications, promising to enhance functional annotations in phages and potentially in other organisms facing complex annotation challenges.

## Supporting information

**S1 Fig. t-SNE representation of the PD domain fold.** ESM-2 embedding representations were computed for each PD domain fold of the training dataset.
(DOCX)

**S1 Table.** (Tab A) List of selected Interproscan entries associated with either E.C.4.4.2 or E.C.3.2.1 and a catalytic activity. (Tab B) Annotation of the proteins used for to filter in the sequences clustered with affinity propagation. (Tab C) Entries PD fold database. (Tab D) Annotation of the proteins used for the negative set of the training data. (Tab E) Metrics of Deposcope on the evaluation dataset across different clustering values. (Tab F) Manually added Triple Helices. (Tab G) Detailed benchmarking results of PhageDPO, DePP and Depo-Scope on the Pires 2016 dataset. (Tab H) Manual analysis of the FPs and FNs in the benchmark.
(XLSX)

**S1 Data. Predicted structures of the FPs and FNs in the benchmark.**
(ZIP)

## Author Contributions

**Conceptualization:** Robby Concha-Eloko, Dimitri Boeckaerts.

**Data curation:** Robby Concha-Eloko.

**Formal analysis:** Robby Concha-Eloko, Dimitri Boeckaerts.

**Funding acquisition:** Bernard De Baets, Yves Briers, Rafael Sanjuán, Pilar Domingo-Calap.

**Methodology:** Robby Concha-Eloko, Dimitri Boeckaerts.

**Project administration:** Dimitri Boeckaerts.

**Software:** Robby Concha-Eloko.

**Supervision:** Michiel Stock, Bernard De Baets, Yves Briers, Rafael Sanjuán, Pilar Domingo-Calap, Dimitri Boeckaerts.

**Validation:** Robby Concha-Eloko, Dimitri Boeckaerts.

**Visualization:** Robby Concha-Eloko, Dimitri Boeckaerts.

**Writing – original draft:** Robby Concha-Eloko, Dimitri Boeckaerts.

**Writing – review & editing:** Robby Concha-Eloko, Michiel Stock, Bernard De Baets, Yves Briers, Rafael Sanjuán, Pilar Domingo-Calap, Dimitri Boeckaerts.

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
