## [Decision Letter · Decision Letter 0]

23 Feb 2024

Dear dr. Boeckaerts,

Thank you very much for submitting your manuscript "DepoScope: accurate phage depolymerase annotation and domain delineation using large language models" for consideration at PLOS Computational Biology.

As with all papers reviewed by the journal, your manuscript was reviewed by members of the editorial board and by several independent reviewers. In light of the reviews (below this email), we would like to invite the resubmission of a significantly-revised version that takes into account the reviewers' comments.

We cannot make any decision about publication until we have seen the revised manuscript and your response to the reviewers' comments. Your revised manuscript is also likely to be sent to reviewers for further evaluation.

Sincerely,

Yang Lu, Ph.D.

Academic Editor

PLOS Computational Biology

Nir Ben-Tal

Section Editor

PLOS Computational Biology

Reviewer's Responses to Questions

**Comments to the Authors:**

Reviewer #1: The review is uploaded as an attachment.

Reviewer #2: ConCha-Eloko et al. developed a language model-based ML method to accurately annotate phage depolymerases and their associated enzymatic domains. To be specific, the authors fine-tuned the language model ESM2 and trained two separate models tasked to predict token classification and binary classification, respectively. The manuscript is well-written overall, but the benchmark experiments have some design flaws. Although, the authors presented some results showing superior performance on binary classification task compared to DePP and PhageDPO on one dataset, the majority of the benchmark analysis is only self-benchmark and lacks baseline and other methods to compare with.

Major comments:

1. The authors mentioned that phage depolymerases have high sequence diversity, but such features were not reflected by DepoScope. For both token and binary experiments shown by table 1, data was split randomly. The good performance using random split could be attributed to the similarity between training and testing set. To rule out similarity element and support the performance of DepoScope, the authors should consider split training and testing using clustering, so that sequences in training and those in test have low sequence similarity. The authors should also consider using cross validation to reduce randomness.

2. For results in table 1, only self-benchmark is shown. The authors should consider including more baselines such as comparing with the traditional bioinformatics methods like BLASTp or using onehot encoding as a baseline to compare with LM embedding.

3. What does “DepoScope correct” mean in table 2?

Minor comments:

1. 2 convolutions followed by two dense layer is rarely called a “decoder” structure, as “decoder” usually implies the upsampling blocks that reconstruct a bottleneck layer.

2. Please double check the citations, for example, Page 19 line 547, the DOI for DePP is wrong.

3. How are prediction cutoff/threshold determined by DePP and PhageDPO on Pires dataset? The precision of both models is awfully low, but recalls are really high, which seems like the cutoff/threshold is too low. Area under precision-recall curve might serve as a better metrics.

Reviewer #3: This paper presents an approach to annotating phage depolymerases by leveraging a fine-tuned ESMFold model, supplemented with two 1D convolutional layers and a dense layer. The writing is well-crafted, and the code is user-friendly. However, it's important to note the presence of established tools for annotating the Enzyme family utilizing deep learning techniques: https://www.ncbi.nlm.nih.gov/pmc/articles/PMC10634757/. The task of annotating phage depolymerases, as described, appears to be more straightforward than the comprehensive annotation required for the Enzyme family, primarily involving binary classification. This simplification could potentially limit the scope of the tool's applicability and innovation. In addition, this paper is lack of downstream analysis, like substrate prediction.

**Have the authors made all data and (if applicable) computational code underlying the findings in their manuscript fully available?**

Reviewer #1: Yes

Reviewer #2: Yes

Reviewer #3: Yes

PLOS authors have the option to publish the peer review history of their article (what does this mean?). If published, this will include your full peer review and any attached files.

Reviewer #1: No

Reviewer #2: No

Reviewer #3: No
---

## [Decision Letter · Decision Letter 1]

21 May 2024

Dear dr. Boeckaerts,

Thank you very much for submitting your manuscript "DepoScope: accurate phage depolymerase annotation and domain delineation using large language models" for consideration at PLOS Computational Biology. As with all papers reviewed by the journal, your manuscript was reviewed by members of the editorial board and by several independent reviewers. The reviewers appreciated the attention to an important topic. Based on the reviews, we are likely to accept this manuscript for publication, providing that you modify the manuscript according to the review recommendations.

Sincerely,

Yang Lu, Ph.D.

Academic Editor

PLOS Computational Biology

Nir Ben-Tal

Section Editor

PLOS Computational Biology

Reviewer's Responses to Questions

**Comments to the Authors:**

Reviewer #1: The authors have effectively addressed the comments. The article can now be considered for acceptance.

Reviewer #2: Overall, the authors have addressed most of my comments. I appreciate the authors' efforts.

For my first major comment, 0.65 sequence identity is still relatively high. Can the authors show the result under 0.3 clustering? Since above 0.3, the sequences can still be considered as homologous.

For my second major comment, I understand that BLASTp may not be the best benchmark to use. However, in authors' response I did not see a strong reason for not using one-hot encoding as a benchmark. Without benchmarking with comparing to one-hot encoding it is hard to claim the superiority of your model is due to ESM2 embedding.

**Have the authors made all data and (if applicable) computational code underlying the findings in their manuscript fully available?**

Reviewer #1: Yes

Reviewer #2: Yes

PLOS authors have the option to publish the peer review history of their article (what does this mean?). If published, this will include your full peer review and any attached files.

Reviewer #1: No

Reviewer #2: No

Figure Files:

Data Requirements:

Reproducibility:

References:

---

## [Decision Letter · Decision Letter 2]

20 Jul 2024

Dear dr. Boeckaerts,

We are pleased to inform you that your manuscript 'DepoScope: accurate phage depolymerase annotation and domain delineation using large language models' has been provisionally accepted for publication in PLOS Computational Biology.

Best regards,

Yang Lu, Ph.D.

Academic Editor

PLOS Computational Biology

Nir Ben-Tal

Section Editor

PLOS Computational Biology

Reviewer's Responses to Questions

**Comments to the Authors:**

Reviewer #1: The paper can be considered for acceptance.

Reviewer #2: Benchmarking using a simple encoder like one hot encoding is a common practice and is often used to justify the choice of a more complex encoder like ESM-2. While I recommend accepting the paper, I reserve my opinion on one hot encoding.

**Have the authors made all data and (if applicable) computational code underlying the findings in their manuscript fully available?**

Reviewer #1: Yes

Reviewer #2: None

PLOS authors have the option to publish the peer review history of their article (what does this mean?). If published, this will include your full peer review and any attached files.

Reviewer #1: No

Reviewer #2: No

---

## [Editor Report · Acceptance letter]

1 Aug 2024

PCOMPBIOL-D-24-00082R2 

DepoScope: accurate phage depolymerase annotation and domain delineation using large language models

Dear Dr Boeckaerts,

I am pleased to inform you that your manuscript has been formally accepted for publication in PLOS Computational Biology. Your manuscript is now with our production department and you will be notified of the publication date in due course.

With kind regards,

Anita Estes
